# Scalable Production and In Vitro Efficacy of Inhaled Erlotinib Nanoemulsion for Enhanced Efficacy in Non-Small Cell Lung Cancer (NSCLC)

**DOI:** 10.3390/pharmaceutics15030996

**Published:** 2023-03-20

**Authors:** Gautam Chauhan, Xuechun Wang, Carol Yousry, Vivek Gupta

**Affiliations:** 1Department of Pharmaceutical Sciences, College of Pharmacy and Health Sciences, St. John’s University, Queens, NY 11439, USA; 2Department of Pharmaceutics and Industrial Pharmacy, Faculty of Pharmacy, Cairo University, Kasr El-Aini, Cairo 11562, Egypt

**Keywords:** hot melt extrusion, nanoemulsion, non-small cell lung cancer, lung delivery, scalability, super-refined excipients, nanorepurposing

## Abstract

Non-small cell lung cancer (NSCLC) is a global concern as one of the leading causes of cancer deaths. The treatment options for NSCLC are limited to systemic chemotherapy, administered either orally or intravenously, with no local chemotherapies to target NSCLC. In this study, we have prepared nanoemulsions of tyrosine kinase inhibitor (TKI), erlotinib, using the single step, continuous manufacturing, and easily scalable hot melt extrusion (HME) technique without additional size reduction step. The formulated nanoemulsions were optimized and evaluated for their physiochemical properties, in vitro aerosol deposition behavior, and therapeutic activity against NSCLC cell lines both in vitro and ex vivo. The optimized nanoemulsion showed suitable aerosolization characteristics for deep lung deposition. The in vitro anti-cancer activity was tested against the NSCLC A549 cell line which exhibited 2.8-fold lower IC_50_ for erlotinib-loaded nanoemulsion, as compared to erlotinib-free solution. Furthermore, ex vivo studies using a 3D spheroid model also revealed higher efficacy of erlotinib-loaded nanoemulsion against NSCLC. Hence, inhalable nanoemulsion can be considered as a potential therapeutic approach for the local lung delivery of erlotinib to NSCLC.

## 1. Introduction

Lung cancer is one of the most common causes of cancer deaths globally with an estimate death of 1.6 million patients each year. Out of these cases, >80% are classified as non-small cell lung cancer (NSCLC) with poor prognosis and 5-year survival rate of only 10% [1]. Currently available therapeutic strategies include immunotherapy, chemotherapy, targeted therapies, and radiotherapy, which usually result in systemic side effects, drug-resistance, and high chances of tumor recurrence and metastasis [2,3]. Ultimately, a lot of patients resort to surgical resection which may damage nearby tissues and organs, causing other infectious diseases such as pneumonia, in addition to its slow recovery [4]. The major drawback of chemotherapy is off-target biodistribution of chemotherapeutic agents in tissues other than tumor site. The need to deliver chemotherapeutic agents directly into the lungs, thus avoiding adverse off-target effects on the non-tumoral tissues can be resolved by developing an inhaled locally accumulating chemotherapeutic dosage form. This can lead to a significant improvement for patients with NSCLC localized in the lungs with no signs of cancer metastasis [5].

Tyrosine kinase inhibitors (TKIs) have been at the forefront of therapeutic advances as targeted therapy against NSCLC [6]. Erlotinib is a first-generation tyrosine kinase inhibitor that acts against NSCLC by inhibiting the function of a protein called epidermal growth factor receptors (EGFR). EGFRs are excessively present on the surface of cancer cells and play a significant role in their division and growth [7]. Erlotinib was approved by the US Food and Drug Administration (FDA) in 2004 as a first-line therapy for NSCLC treatment, and was further extended to treat advanced metastatic pancreatic cancer in 2016 [8]. Despite its pharmacological benefits, erlotinib is a BCS Class II drug (low solubility and high permeability), which may be responsible for well-documented concerns regarding its poor bioavailability [9]. Currently, it is only available for oral administration, which undergoes an extensive metabolism resulting in moderate bioavailability (≈60%), dose inadequacy, and adverse side effects [10]. Thus, developing an inhaled erlotinib delivery system is necessary for the direct targeting of NSCLC in the lungs with adequate doses. Previous studies from our and other research groups have reported the formulation of inhaled erlotinib through cyclodextrin complexation [11], encapsulation into PLGA nanoparticles [7], lipid nanoparticles [12,13], and liposomes [14], as well as the development of dry powder inhalers [15,16]. All these formulations were prepared using bench-top batch techniques such as emulsification, ultrasonication, homogenization, etc., which are hard to scale up, and subsequently, difficult to adapt to pharmaceutical industry. Other disadvantages for these batch techniques include time consumption, cost, labor, and inter-batch variability. On the other hand, pharmaceutical industry adopts continuous manufacturing techniques, which are easy to translate with low cost, labor, and space. Continuous manufacturing approaches for pharmaceuticals are scalable and high throughput, and have been repeatedly vetted by the FDA for utilization in drug product manufacturing [17,18]. In addition, these techniques can be operated with Realtime analytics, thus providing insights on the product quality for better decision making and continuous quality improvement. In this article, we have used the hot melt extrusion (HME) method, a well-established continuous manufacturing technique, to fabricate an inhaled erlotinib dosage form.

Hot melt extrusion (HME) is a continuous manufacturing technique which has been used to prepare numerous FDA-approved products on the market [19]. In this technique, the material is passed through an extruder under high shear and temperature to give a homogenously dispersed product. It is a well-explored technique used to prepare various dosage forms, including tablets, films, suspensions, emulsions, and nano- and microparticles [20,21]. HME is an easy to scale up technique, as it functions on independent modular parameters, such as melting temperature, screw configuration, and screw speed, which do not require large modifications to scale up from a mini extruder (5–10 g/h output) to an industrial scale extruder (10–100 kg/h output). Furthermore, with the help of techniques such as Raman spectrophotometer, the material processed through HME can be analyzed in real time for quality control. Additionally, unlike other conventional techniques used to prepare nanoemulsion, HME does not require organic solvents to dissolve the excipients and drugs, as they simply melt at high temperature and pressure to form nanoemulsions. HME has been traditionally used to prepare solid dosage form by extruding drugs along with thermoplastic polymers, which require a high processing temperature. With the help of lipids, the processing temperature is easily reduced to improve the drug selectivity and processability [22,23]. Some of the common lipids used for HME include Glyceryl behenate (Compritol^®^ 888 ATO), castor oil, soybean oil, palmitic acid/stearic acid, polyoxyethylene hydrogenated castor oil (Kolliphor^®^ RH 40), vitamin E TPGS, polysorbate (Tween), glycerol monostearate/butyl stearate, glyceryl palmitostearate (Precirol^®^ ATO 5), lauroyl polyoxylglycerides (Gelucire^®^ 44/14), stearoyl polyoxylglycerides (Gelucire^®^ 50/13), and PEG-8 caprylic-capric glycerides (Labrasol^®^) [24,25,26,27,28]. In this study, we have prepared nanoemulsion using Super Refined™ excipients which are highly purified and free of primary or secondary oxidative impurities such as peroxides, aldehydes, ketones, and catalyst residues, which can be used to improve the stability of the product after exposing to heat stress via HME [29].

While HME has been well established in pharmaceutical industry, its applications in developing drug-encapsulated nanocarriers remains relatively unexplored. A few published reports have explored use of HME, followed by secondary size reduction techniques, such as high-pressure homogenizer and ultrasonication, in nanoparticle preparation [30,31]. However, to the best of our knowledge, this study is the first to report the formulation of an inhaled drug-loaded lipid nanoemulsion via single-step hot melt extrusion technique using super refined excipients.

In this study, we prepared inhaled erlotinib nanoemulsion using a scalable HME technique and tested the dosage form for physiochemical properties, in vitro pulmonary deposition behavior and therapeutic activity against NSCLC cell lines. We began with screening Super Refined™ excipients for maximum drug solubility. The selected drug–excipient mixtures were then processed through HME, as shown in schematics Figure 1, while adding water directly into the extruder allowing high shear mixing of lipid and water to form nanoemulsions. Thereafter, we examined the capability of scalable erlotinib nanoemulsions to carry the drug to the respirable region of the lungs, thus reducing the dose and off-target effects, preferentially targeting cancer cells for therapeutic activity.

## 2. Materials and Cell Lines

Lipids and solubilizers used in this study include TWEEN 80 HP-LQ-(MH), super refined Propylene glycol-LQ-(MH), and super refined L18 POG (Polyethylene Glycol), which were received as investigational samples from CRODA, Inc. (Princeton, NJ, USA). Erlotinib Hydrochloride (Erlo) (>99% pure) was purchased from LC laboratories (Woburn, MA, USA). Other chemicals, such as sodium chloride, potassium chloride, potassium phosphate, and O-phosphoric acid, were purchased from Fisher Scientific (Hampton, NH, USA). All other reagents and solvents were of analytical grades and were purchased from third party vendors.

A549 NSCLC cell lines were obtained from ATCC (Manassas, VA, USA) and maintained in RPMI-1640 medium (Corning, NY, USA) supplemented with 10% FBS (R&D Systems, Minneapolis, MN, USA), sodium pyruvate (1% *v*/*v*), and penicillin–streptomycin (1% *v*/*v*) (Corning, NY, USA), incubated at 37 °C/5% CO_2_, and cultured to 85–90% confluency.

## 3. Methods

### 3.1. Experimental Setup

The nanoemulsions were prepared using a co-rotating twin screw extruder (HME; Process 11 Extruder, Thermo Scientific, Bridgewater, NJ, USA). The extruder consists of three major parts, the motor, which acts as a drive unit, a temperature barrel, which consists of eight heating zones maintained at specific temperature with the help of recirculating ambient water, and twin screws, which are composed of conveying screws and three sections of kneading screws for pushing the material through the extruder under high shear mixing. The HME setup for this experiment was modified with water addition ports at zones 2 and 4 adjacent to the mixing screws for complete homogenization of the lipids and water together. The temperature of all the zones were kept above the temperature at which the drug solubilizes into the lipid and form homogeneous mixture under high temperature and pressure. The extruded emulsion was collected in a container and stored for further characterization.

### 3.2. UPLC Method for Erlotinib

A reverse-phase liquid chromatography method was developed to quantitively analyze erlotinib using Waters^®^ Acquity series UPLC (Waters, Milford, MA, USA). The method was developed using an analytical column C18–2.5 µm particles and 3.0 × 50 mm dimension (Waters Acquity, Milford, MA, USA) with column temperature maintained at 25 °C. The mobile phase consisted of an acetonitrile and 0.1% orthophosphoric acid ratio of 85:15, with a flow rate of 0.6 mL/min and UV detection at a wavelength of 247 nm. The chromatogram was obtained and analyzed using Empower 3 software (Waters).

### 3.3. Preliminary Screening of Lipids and Solubilizers

#### 3.3.1. Drug–Excipient Solubility

The preliminary screening was carried out using Super refined grades of lipids and solubilizer provided by CRODA, Inc. (Princeton, NJ, USA), including Tween 80, propylene glycol, and L18 POG. The drug solubility was examined by adding an excess amount of drug (~50 mg) in 1 mL of these excipients as well as their binary and tertiary mixtures, as described in Table 1. The drug–excipient mixtures were then stirred in a tube rotator overnight at room temperature. The next day, the tubes were centrifuged (Eppendorf, Framingham, MA, USA) at 21,000× *g* for 30 min to remove the undissolved drug and the supernatant was analyzed for drug content using the UPLC method mentioned above. The excipient mixture with higher drug content were further screened for drug solubility at a higher temperature. The drug–excipient mixtures were heated at 70 °C in a water bath for 3 h with occasional vortexing at high speed. Thereafter, the tubes were centrifuged and analyzed for drug content, as previously mentioned. From these studies, formulations E5 and E6 were selected for further studies.

#### 3.3.2. Dispersibility of the Nanoemulsion

Before finalizing the drug–excipient mixture to be used for HME, the effect of water addition on the stability of the promising nanoemulsions, E5 and E6, was assessed. This was conducted using a previously reported method by increasing the water ratio in the drug–excipient mixture from 1:1 to 1:2 and 1:3 [32,33]. Thereafter, the tubes were heated to 70 °C in a water bath, vortexed to form an emulsion, and then left to equilibrate for 24 h before visually analyzing the emulsion for any sign of breakage, creaming, flocculation, etc. [33,34].

### 3.4. Nanoemulsion Preparation by HME

For preparing nanoemulsions using optimized drug/excipient/water mixture, the drug–excipient mixture was added to the extruder through the hopper connected to the barrel, with the barrel temperature maintained at 80 °C to completely dissolve the drug into the excipient. The mixture was then allowed to pass through the extruder at high screw speed of 300 RPM with a torque of <10%. Specific amount of water was added to the extruder at Zones 2 and 4, adjacent to the kneading element to ensure the homogenous mixing of the excipients under high temperature. The run time for the material through the extruder was just 30–35 s, where ~10–15 mL of the hot emulsion was collected from the barrel’s discharge end. The collected emulsion was then centrifuged at 5000 rpm for 5 min to remove any debris or undissolved drug from the final nanoemulsion. The prepared nanoemulsions were analyzed for particle characteristics using DLS Zetasizer (Malvern Panalytical Ltd., Westborough, MA, USA) and drug content using UPLC, following the method described in Section 3.2.

### 3.5. Solid-State Characterization

#### 3.5.1. Diffraction Scanning Calorimeter (DSC)

The DSC studies for erlotinib, the excipient mixtures and the prepared nanoemulsion were performed using the Perkin Elmer DSC 6000 (PerkinElmer, Waltham, MA, USA). The formulations were first lyophilized using FreeZone Benchtop Freeze Dry system (LabConco, Kansas City, MO, USA). Thereafter, 6–7 mg of the lyophilized formulation was put in an aluminum pan using a micropipette and hermetically sealed. The sealed pan was then carefully placed on the microscale inside the sample chamber, which was first equilibrated at 30 °C for 5 min and then the temperature was increased from 30 °C to 300 °C at a heating rate of 10 °C/min. The heat flow was recorded and plotted against the temperature. The data were then analyzed using the TA instruments universal analysis 2000 software.

#### 3.5.2. Fourier Transform Infrared Spectroscopy (FT-IR)

FT-IR analysis for erlotinib, the excipient mixtures, and the prepared nanoemulsion were carried out using Spectrum-100 (PerkinElmer, Waltham, MA, USA), equipped with attenuated transmittance reflectance (ATR), where water was used for background correction. The samples were carefully placed on the diamond crystal and analyzed over a wavenumber range of 700 to 4000 cm^−1^.

### 3.6. Transmission Electron Microscopy (TEM) Imaging

The morphology of the nanoemulsion droplets was studied using TEM. Briefly, 5 µL of the sample was placed on a formvar-carbon-coated copper grid (100 mesh, Electron microscopy sciences, Hatfield, PA, USA). Thereafter, the grid was rinsed twice with milli Q water and stained with 2% uranyl acetate solution (Ladd Research Industries, Williston, VT, USA), and any excess solution was removed with Whatman 3MM filter paper. The grid was then left to air-dry for a few minutes and imaged at 20,000× magnification using FEI Tecnai Spirit TWIN TEM (FEI, Hillsboro, OR, USA) operated at 120 kV voltage.

### 3.7. In Vitro Release of Erlotinib from Nanoemulsions

In vitro release study was executed to determine the time-dependent release of erlotinib from the prepared nanoemulsion using dialysis cassette method. Briefly, a 10,000 MWCO dialysis cassette (Slide-A-Lyzer, Thermo Scientific, Waltham, MA, USA) was hydrated with 0.1% Tween 80 solution. Thereafter, 0.5 mL of the formulation was added to the dialysis cassette using a BD syringe with a 19G1_1/2_ TW needle and placed inside the release medium, comprising 100 mL of 0.1% Tween in phosphate buffer saline (PBS) maintained at 37 ± 0.5 °C while stirring at 100 RPM. The release samples were collected at time points, 0.25, 0.5, 1, 2, 4, 6, 8, 12, 24, 72, and 144 h, while replacing collected volume with fresh media. The collected samples were diluted with the same volume of Acetonitrile and analyzed using the UPLC method described in Section 3.2. The release kinetics and pattern of drug release from the nanoemulsion was determined by fitting the release data to distinct models; zero-order model, first-order model, Higuchi model, Korsmeyer–Peppas model, and Hixson–Crowell model.

### 3.8. Stability Studies

The stability study for the prepared nanoemulsion was carried out in accelerated conditions, at 40 °C and 75% relative humidity. The liquid nanoemulsion samples were stored in the stability chamber for a period of 4 weeks. At the end of incubation period, samples were analyzed for particle characteristics, drug content, and physiochemical characteristics, as previously discussed.

### 3.9. Inhalability, Aerosolization, and Aerodynamic Properties

In vitro aerosolization study in the lungs was assessed using Next-Generation Impactor (NGI; Model 170, NGI: MSP Corp., Shoreview, MN, USA). NGI can collect aerosol droplets at different stages based on their aerodynamic size, which is useful in predicting clinical lung deposition of the inhaled dosage formulations. Briefly, 2 mL of the formulation was loaded in the nebulizer cup attached to PARI^®^ LC PLUS Nebulizer, which was then drawn through the NGI with the help of a vacuum pump (Copley HCP5) operated at 15 L/min for 4 min. The droplets were collected from each stage and drug content was analyzed using the developed UPLC method. The fine particle fraction (FPF, %) was determined as the fraction of emitted dose deposited in the NGI with Aerodynamic diameter (d_ae_) < 5.39 µm or amount of drug deposited from stage 3–8 to the total emitted dose. The mass median aerodynamic diameter (MMAD, D_50%_) was obtained by determining the aerodynamic diameter that corresponds to 50% of the total cumulative deposition. The calculation for MMAD (D50%) and GSD were done using an earlier reported method [35].

### 3.10. In Vitro Cytotoxicity

Cytotoxicity studies were performed in A549 NSCLC cells to confirm the therapeutic efficacy of the prepared nanoemulsion. The cells were sub-cultured in 96-well plate at a density of 2500 cells/well and incubated for 12 h at 37 °C/5% CO_2_. The cells were then treated with the plain drug and the prepared nanoemulsion at a concentration ranging from 0.09 to 12.5 µM along with the no treatment media as control, followed by incubation over a period of 72 h at 37 °C/5% CO_2_. The cellular viability was then evaluated using an MTT assay using a Tecan Spark 10M Plate reader (Tecan, Männedorf, Switzerland), following protocols established earlier by our group [36]. All cytotoxicity studies were repeated at least three times with n = 6 each time. The IC_50_ values were calculated using the non-linear fitting module in GraphPad Prism 6.0 Software (GraphPad Software, San Diego, CA, USA). The cytotoxicity of the blank nanoemulsions without the drug was also tested to ascertain the safety of excipients without any byproducts or contaminants formed upon exposure to high temperature and pressure in HME.

### 3.11. In Vitro 3D Tumor Simulation Studies

To further study the efficacy of prepared nanoemulsion against NSCLC, we carried out the in vitro tumor simulation studies using 3D spheroids, which not only determine the cytotoxic potential, but also the penetrability of the formulation into the solid tumor mass. Our group has earlier established feasibility of A549 cells in forming sizable 3D spheroids [37,38]. Briefly, A549 cells were plated in an ultra-low attachment U-shaped cell culture plate at a density of 2000 cells/well. Cells were incubated at 37 °C/5% CO_2_ for 3 days (72 h) to grow and form spheroids, following which the spheroids were treated with either a single dose or multiple doses of erlotinib-free solution and erlotinib-loaded nanoemulsions at two different concentrations, 4 and 8 µM, against blank (drug-free) media as the control. For the single dose, the media was replenished after every 72 h by replacing half of the media from the wells with fresh media. For multiple doses, half of the media was replenished with 8 and 16 µM concentration of the corresponding treatment to attain the original concentration of 4 and 8 µM, respectively. The images were captured using LAXCO LMI-6000 series inverted microscope (LAXCO, Mill Creek, WA, USA) at 10× magnification on day 3, 6, 9, 12, and 15 post treatment. At the end of the treatment period, 3D spheroid tumor cell viability was measured using cell titer-glo^®^ 3D (Promega Inc., Madison, WI, USA), as per the manufacturer’s protocol.

### 3.12. Live/Dead Cellular Assay

Live/Dead cell assay was performed using Viability/Cytotoxicity Assay Kit for Animal Live and Dead Cells (Biotium, Fremont, CA, USA) on the 15th day of the spheroid study for both single and multiple treatment groups according to the manufacturer’s protocol. The assay provides green fluorescent by calcein AM for staining live cells and red fluorescence by ethidium homodimer III (EthD-III) staining the dead cells. The images were captured by an Evos FL fluorescence microscope at 10× magnification (Thermo Fisher Scientific, Waltham, MA, USA) using GFP (green fluorescence protein) and RFP (red fluorescence protein) filters, respectively. The images were analyzed for fluorescence intensity using ImageJ 1.42 software.

### 3.13. Data Representation and Statistical Analyses

All data were presented as mean ± SD or mean ± SEM, with at least n = 3, unless mentioned otherwise. Statistical analyses were performed with GraphPad Prism 6.01 (San Diego, CA, USA) using Student’s *t*-test, one-way ANOVA, and Tukey’s post-hoc multiple comparison. Statistical significance was considered at *p <* 0.05, presented in data figures as a single asterisk (*). However, some studies demonstrated a smaller *p*-value of 0.01 or less, which is indicated at respective places.

## 4. Results

### 4.1. UPLC Method

The analytical method for analyzing erlotinib was successfully established on the Waters Acquity UPLC using the method described in Section 3.2. A sharp quantitative peak eluted at a retention time of 0.773 min, shown in Appendix A.

### 4.2. Preliminary Screening

The preliminary screening was performed by quantifying the solubility of erlotinib in L18 POG, propylene glycol and Tween 80, as well as in their binary and ternary mixtures (Table 1). The results showed the highest solubility in Tween 80 at concentration of 7.3 ± 0.9 mg/mL, whereas the solubility was reduced to 5.1 ± 0.6 and 3.3 ± 0.4 mg/mL in combination with propylene glycol (E5) and L18 POG (E6), respectively. This showed that erlotinib has high affinity for Tween 80, which was selected for further evaluation in combination with the other excipients for thermal solubility study. Henceforward, the prepared drug–excipient mixtures were heated at 70 °C and regularly vortexed over a period of 3 h. As presented in Table 2, the results showed an increase in drug solubility from 5.1 mg/mL to 8.0 ± 0.2 mg/mL for propylene glycol and Tween 80 mixture (E5), even though the solubility observed for L18 POG and Tween 80 (E6) was almost unchanged (3.5 ± 0.1 mg/mL).

The drug–excipient mixtures prepared at elevated temperature are supersaturated systems. Therefore, drug precipitation could be a major stability concern for the prepared nanoemulsions. Thus, A 24-h stability analysis was performed where the emulsions prepared using a mixture of drug, excipients, and water mixed at 70 °C were left to equilibrate for 24 h at room temperature. Visual observation of the different formulations revealed that the E6 prepared using L18 POG and Tween 80 resulted in phase separation at different lipid:water ratios due to creaming of the emulsion. In contrast, the emulsion prepared using propylene glycol and Tween 80 (E5) were stable with no phase separation or drug precipitation issues (Table 3). Representative images are also shown in Table 3. Accordingly, E5 was selected for HME at 1:1 (E5-1) and 1:2 (E5-2) ratios of lipid:water.

### 4.3. Nanoemulsion Preparation by HME

The nanoemulsions were prepared using a single-step HME, as shown in Table 4. E5-1 prepared using a 1:1 ratio of lipid:water showed erlotinib content of 4.8 ± 0.7 mg/mL (55.8 ± 5.7% drug entrapment), globule size of 186.4 ± 97.6 nm, polydispersity index (PDI) of 0.23 ± 0.03, and zeta potential of −3.2 ± 5.1 mV. Increasing the amount of water to 1:2 (E5-2) showed an erlotinib concentration of 4.4 ± 1.0 mg/mL (66.7 ± 15.7% drug entrapment), with 122.2 ± 58.6 nm globule size, 0.36 ± 0.14 PDI, and zeta potential of −0.7 ± 1.6 mV. There was no significant difference observed in the drug content and globule size for E5-1 and E5-2. However, the magnitude of zeta potential for E5-1 was slightly lower than E5-2, and the water content for E5-1 is less than E5-2, which can provide better stability. Therefore, E5-1 was selected for further evaluation.

### 4.4. Solid-State Characterization

To determine the complete dissolution of the drug within the formulated nanoemulsion, DSC was performed. As denoted in Figure 2A, the DSC of erlotinib shows a sharp endothermic peak at 172 °C that corresponds to its melting temperature and indicates the crystalline nature of the drug. Similar sharp endothermic peak can be observed in the drug excipient mixture before extrusion, which suggests the crystallinity of the drug within the drug–excipient mixture. However, this characteristic peak for erlotinib was not recorded in the final formulation after extrusion, which suggests that the prepared nanoemulsions do not have any drug precipitates in the crystalline state.

To further understand the drug–excipient chemical interactions inside the final formulation, we carried out an FT-IR study. As shown in the FT-IR spectra presented in Figure 2B, erlotinib spectrum had signal peaks at 2900, 2875, 1625, 1450, and 1120 cm^−1^, corresponding to CH stretching, CH stretching, NH bending, C=C aromatic stretching, and C-OH stretching, respectively, with a broad band at 3350 cm^−1^ attributed to O-H stretching due to the presence of water. Similar significant peaks at 3392, 2900, 2874, 1625, 1450, and 1118 cm^−1^, corresponding to the functional groups of the drug, were also detected in the spectrum for the drug–excipient mixture and the final nanoemulsion [39,40]. These results demonstrate no covalent bonding between the excipients and drug in the E5-1 nanoemulsion [41,42,43].

### 4.5. TEM Imaging

The nanoemulsion droplets were analyzed for morphology using TEM imaging and a representative image is presented in Figure 2C. The images display emulsion droplets with globule size ranging from 150–200 nm, which is in accordance with the globule size attained from DLS Zetasizer measurement.

### 4.6. In Vitro Release

The in vitro release studies were completed in 0.1% Tween 80 PBS; the release profile is presented in Figure 3A, which shows continuous sustained release of erlotinib over a period of 6 days, with only 4% drug release in the first 2 h, thus highlighting minimal burst release. The drug release was observed to be time dependent, with 20.6 ± 0.3% release at 24 h, 43.2 ± 3.4% at 72 h, and 61.5 ± 2.8% erlotinib released at the end of 144 h (6 days). Thereafter, the release data were fitted into various release models, i.e., zero-order, first-order, Hixson–Crowell, Kosmeyer–Peppas, and Higuchi models, to understand the release kinetics of the formulation (Figure 3B–F), with r^2^ values listed within the graphs. As can be seen, the Higuchi model was found to be the best fitting model with the highest correlation coefficient value (R^2^) of 0.9993, suggesting that erlotinib release from the nanoemulsion droplets follows the diffusion-controlled mechanism.

### 4.7. Stability

The stability of the formulated nanoemulsion E5-1 was investigated at accelerated conditions of 75% RH and 40 °C. As presented, Figure 4A(i) shows no significant change in the amount of drug in the nanoemulsion represented as the percentage of drug entrapment: 45.4 ± 1.6% and 39.1 ± 11.8% for week 0 and week 4, respectively. There was a small increase in the particle size from 122.2 ± 6.1 nm to 173.7 ± 38.9 nm, with no unusual change in the PDI (Figure 4A(ii,iii)). However, a significant change in the zeta potential value from 1.6 ± 2.3 mV at week 0 to 8.4 ± 2.6 mV at week 4 was observed (Figure 4A(iv)). On the other hand, the DSC chromatograms for the formulated nanoemulsion did not display the characteristic sharp endothermic peak of erlotinib at 172 °C either at week 0 or after 4 weeks (Figure 4B). This indicates that the drug did not recrystallize and remained in the amorphous state inside the formulation during the storage period.

### 4.8. Inhalability, Aerosolization, and Aerodynamic Properties

The deposition of nanoemulsion inside the lungs depends on multiple factors, such as droplet size, shape, density, and surface charge, along with other anatomical and physiological barriers of the lungs. The aerodynamic properties of the nebulized nanoemulsion droplets were estimated by calculating the mass median aerodynamic diameter (MMAD), which describes the median aerodynamic particle size distribution of the aerosol by mass, GSD, which defines the spread of the aerodynamic particle distribution, and the percentage of fine particle fraction (%FPF), which is the fraction of the emitted dose that will reach the deep lung region. The results are presented in Figure 5A, with MMAD as 3.9 ± 0.2 µm, GSD up to 2.6 ± 0.1 µm, and %FPF of 76.2 ± 0.2%, suggesting that most of the nanoemulsion nebulized using a jet nebulizer reached the respirable region of the lungs. In addition, the deposition profile of the aerosol representing the percentage of deposition respective to each stage, presented in Figure 5B,C, shows maximum deposition at stage 4 and 5, which correspond to deep lungs. These results indicate good aerosolization performance of the nanoemulsions and ensure their inhalability.

### 4.9. In Vitro Cytotoxicity

The cytotoxicity of E5-1 nanoemulsion was estimated by measuring the percentage of cell viability of erlotinib-resistant NSCLC A549 cells using the MTT assay compared to the erlotinib-free solution. The cells were treated with different concentrations and cell viability was estimated after 72 h. As shown in Figure 6A, both erlotinib and E5-1 demonstrated concentration-dependent cytotoxicity, with E5-1 showing better cell killing efficacy compared to the plain erlotinib (Figure 6A). As can be seen, E5-1 nanoemulsion demonstrated significantly lower IC_50_ of 2.7 ± 2.1 µM compared to 7.7 ± 3.6 µM observed in the erlotinib-free solution (*p <* 0.05). These results suggest an enhanced cytotoxicity of the E5-1 nanoemulsion against NSCLC cells and potential clinical dose reduction following encapsulation in lipid-based nanoemulsions. Drug-free nanoemulsion formulation was also tested for cytotoxicity in NSCLC A549 cells to ensure that the HME procedure performed under high temperature and pressure did not result in toxic byproducts. As observed, the blank nanoemulsion resulted in high cell viability (>90%) at all concentration levels up to 12.5 µM (Figure 6B). The enhanced cytotoxic effect of E5-1 nanoemulsion against NSCLC can be attributed to improved nanoemulsion-medicated drug accumulation inside the cells as lipid and surfactant components of the nanoemulsion can improve cell membrane mobility, and subsequently, enhanced therapeutic activity [44].

### 4.10. In Vitro 3D Tumor Simulation Studies

In vitro cytotoxicity in a 2D model with cells attached on the plate surface does not mimic a solid tumor mass growing in all directions inside the lungs; therefore, A549 cells were utilized to form compact 3D spheroids to analyze the treatment efficacy of the E5-1 nanoemulsion. In recent years, our and other research groups have shown the impact and utility of 3D spheroid studies in determining efficacy of nanoencapsulated dosage forms [45,46,47]. In both single- and multiple-dose studies, a tight tumor mass was formed after incubating for 3 days. As can be seen in the representative images shown in Figure 7A, after 15 days, a reduction in tumor volume for both 4 µM and 8 µM E5-1 treatment was observed that was visibly more significant compared to erlotinib-free solution. The spheroid volume was quantified using ImageJ software and normalized to the volume of control (no treatment) spheroid. The spheroid volume for E5-1 decreased to a factor of 0.5 ± 0.12 at 4 µM and 0.4 ± 0.1 at 8 µM, as compared to plain drug 1.0 ± 0.2 at 4 µM and 0.7 ± 0.0 for 8 µM (Figure 7B).

In another set of experiments, spheroids were treated multiple times (every 3 days) with respective treatments at 4 µM and 8 µM erlotinib equivalent concentrations. As seen from representative images shown in Figure 8A, a more robust reduction in tumor spheroid volume was observed, especially with E5-1 treatment groups. Following quantification, a similar decrease in the spheroid volume was detected with multiple-dose treatment, as shown in Figure 8B, to a factor of 0.5 ± 0.1 (4 µM) and 0.2 ± 0.0 (8 µM) for E5-1 nanoemulsion as compared to plain drug erlotinib, which showed values of 1.2 ± 0.2 (4 µM) and 1.2 ± 0.2 (8 µM), while the control had values of 1.3 ± 0.1. These results clearly point toward the superior therapeutic efficacy of E5-1 nanoemulsion, as indicated by the significant reduction in the spheroid volume when compared to erlotinib-free solution.

### 4.11. Live/Dead Cellular Assay and 3D Spheroid Cell Viability Studies

The previous efficacy study was based on the physical appearance and volume of the spheroids, which does not give a complete illustration of the therapy as dead cells, which cannot be identified optically, may be present inside the spheroid mass. Thus, we decided to estimate the proportion of cells alive and dead using a live/dead assay which distinguishes two fluorescent dyes, i.e., green (Calcein AM, to stain live cells) and red (EthD-III, to stain dead cells). The fluorescent images were taken for both treatment groups, presented in Figure 9A, and RFP intensity signifying the dead cell population were then calculated per mm^2^ of spheroid surface area, normalized to control group, and plotted for single-dose treatment (Figure 9B) and multiple-dose treatment (Figure 9C). As can be seen from representative images in Figure 9A, GFP intensity decreased significantly for both E5-1 treatment groups (compared to both control and plain erlotinib) in both single and multiple dose treatment (Figure 9A). Upon quantification, a significant increase in RFP intensity was observed for E5-1 treatment groups, outlining a higher number of dead cells. The results of the live/dead assay were in agreement with the previously discussed visual assessment, where E5-1 nanoemulsion showed the highest proportion of dead cells represented as RFP intensity of 208.4 ± 13.1 (compared to 119.4 ± 0.00 for plain erlotinib, 4 µM, *p <* 0.001) and 192.7 ± 29.7 (compared to 134.9 ± 30.7 for plain erlotinib, 8 µM, *p <* 0.05) for the single dose (Figure 9B). The RFP intensity furtherly increased for E5-1 nanoemulsion formulation after multiple doses to 292.9 ± 40.6 (compared to 187.1 ± 34.9 for plain erlotinib, 4 µM, *p <* 0.01) and 252.3 ± 54.4 (compared to 176.5 ± 27.9 for plain erlotinib, 8 µM, ns) (Figure 9C). These results suggest the good spheroid penetrability of the prepared E5-1 nanoemulsion relative to erlotinib solution.

Additionally, the percentage of cell viability for the spheroids using Cell Titer Glo^®^ 3D cell viability assay was performed. Results presented in Figure 10A (single dose) and Figure 10B (multiple dose) showed a significantly lower cell viability with the spheroids treated with E5-1 nanoemulsion as compared to spheroids treated with erlotinib-free solution (considering the control group as 100% cell viability). A single 4 µM and 8 µM dose of erlotinib-free solution showed respective percentages of cell viability of 91.1 ± 1.0% and 87.7 ± 1.7%, which were reduced to 73.7 ± 2.2% and 61.6 ± 5.5% for spheroids treated with E5-1 nanoemulsion (*p <* 0.0001). Similarly, for multiple doses, the percentages of cell viability of spheroids treated with erlotinib-free solution were 58.7 ± 2.9% (4 µM) and 42.6 ± 5.9% (8 µM) and significantly decreased to 36.7 ± 4.8% (4 µM, *p <* 0.001) and 1.8 ± 0.3% (8 µM, *p <* 0.0001) for E5-1 nanoemulsion. These results demonstrate the high efficacy of the prepared E5-1 nanoemulsion compared to drug-free solution in inhibiting 3D spheroid growth over a treatment period of 15 days.

## 5. Discussion

Non-small cell lung cancer (NSCLC) is the most common type of lung cancer that is usually caused by continuous exposure to metals, asbestos, minerals, smoke, air pollution, etc. Although NSCLC grows slower than small cell lung cancer phenotype, it is more likely to metastasize to other body organs, rendering it a life threatening condition, and therefore, early detection and treatment is imperative for improved patient outcomes [48]. One of the most promising target therapies for NSCLC is Tyrosine kinase inhibitor agents (TKI), including erlotinib, gefitinib, osimertinib, and crizotinib [49]. Erlotinib and other TKIs are used to treat metastatic NSCLC in patients with abnormal EGFR gene mutations [50]. In a recent study, they were given to NSCLC patients in the ICU and was found to have better survival rate with decreased risk of acute respiratory failure [51]. While currently available as an oral dosage form, a pulmonary route of delivery could be more suitable for NSCLC treatment, as it offers direct deposition of drug at the target site with high local concentration, results in fewer systemic side effects, and avoids first pass metabolism. Therefore, several attempts have been made to formulate pulmonary delivery systems for erlotinib using solid lipid nanoparticles and liposomes prepared using the thin-film hydration and hot homogenization methods [52,53]. These formulations showed a good inhalation ability, but the applied procedures are bench-top, multi-step techniques that are hard to implement for large-scale production, rendering them unfeasible. In this study, we prepared an inhaled erlotinib nanoemulsion formulation using hot melt extrusion, a continuous manufacturing technique which can be easily scaled up to large scale production in pharmaceutical industry. While a few studies were reported for preparing lipid-based formulations using hot melt extrusion, most of them usually require an additional size reduction step, such as high-pressure homogenization or sonication [19,30]. In our study, we were able to prepare lipid-based nanoemulsions using HME without any additional steps using super refined grade excipients provided by CRODA Inc. The provided high-purity excipients have been processed to remove any impurities, including primary and secondary oxidation products for better stability and formulation integrity [54,55].

The solubility studies showed highest drug solubility with Tween 80, a surfactant with high HLB value (>12), and the hydrophobic co-solvent, propylene glycol, with a low HLB value (<3) [56]. This combination of a hydrophilic surfactant and a hydrophobic co-solvent resulted in high drug solubilization capacity because of the low log *p*-value of erlotinib (Log *p* < 3.5) [57,58]. Therefore, these excipients were used to dissolve the drug and process through HME with continuous addition of water to form nanoemulsion. After adding water, we selected the drug–excipient mixture with higher drug load, lowest particle size, and preferably lower water content to avoid any stability concerns due to the higher water content, as considerable drug precipitation might occur on further dilution [59,60]. The nanoemulsion with the highest drug loading efficacy and smallest particle size was selected to achieve the required therapeutic effect to avoid the undesirable side effects related to excess excipients. Thus, E5-1 nanoemulsion formulation was selected to test for efficacy against NSCLC and suitability for non-invasive lung delivery.

The solid state of the drug inside the nanoemulsion was assessed using DSC, which suggests complete solubilization of the drug within the globules with no drug-free precipitates. In addition, the FTIR study demonstrated the absence of any drug–excipient covalent linkage or any chemical change in the drug structure as all the distinct peaks for erlotinib were present in the drug–excipient mixture and the nanoemulsion E5-1, without any change in their frequency position. The nanoemulsion droplets were also analyzed using TEM, which showed nanoglobules of drug loaded nanoemulsion with globule size ranging from 100–200 nm. The in vitro drug release study was performed in 0.1% Tween 80–PBS solution to simulate the surfactant concentration in lung fluid [61]; the nanoemulsion showed a sustained release profile following a diffusion-based release mechanism in which the drug nanocrystals, which is completely solubilized within the oil phase, is slowly released into the external release medium.

The stability of nanoemulsion is usually a major concern due to the thermodynamic instability caused by the higher free energy state of the emulsion than the free oil and water phase [62,63]. The accelerated stability studies for the formulated nanoemulsion showed no significant difference in the formulation characteristics and drug entrapment efficacy, except for the zeta potential, which increased multiple folds, giving a rather more stable nanoemulsion with lower chances of globules aggregation due to the repulsive positive surface charge [64,65]. In addition, physiochemical characterization via DSC further demonstrates the stability of the nanoemulsion and the absence of any drug precipitation. This can be attributed to the use of super refined excipients, which are highly pure and induce lower oxidative stress, thus providing better stability and formulation integrity [54,55].

The pulmonary route of administration helps to deliver the dose in proximity of the tumor inside the lungs with reduced exposure to other organs, and subsequently, lower adverse side effects. Therefore, an in vitro aerosolization study was conducted to ensure the inhalability of the formulated nanoemulsions in which the aerodynamic properties, including MMAD and FPF of the nebulized nanoemulsion, were found suitable for inhalation. To summarize, the formulated nanoemulsion attained high drug loading with sustained drug release and high inhalability, which can help in reducing the dose size and frequency, as the nebulized droplets will retain the drug inside the lungs, having a long therapeutic efficacy with a single dose.

Thereafter, we tested the in vitro efficacy of the nanoemulsion against NSCLC; the A549 cell line was selected, as it is widely used to study the activity against NSCLC. The cytotoxicity study revealed a much lower IC_50_ value for the E5-1 nanoemulsions as compared to the plain drug, which suggests better internalization of the drug into the cells using the nanoemulsions. On the other hand, the drug-free nanoemulsion showed high cell viability without any toxic effects on the cells, which ascertained the safety of the excipients and proved that the enhanced cytotoxicity of E5-1 nanoemulsion is attributed mainly to the enhanced cellular uptake of the drug.

The 3D in vitro model study is necessary to understand the efficacy of the nanoemulsion in a real tumor-mimicking environment against the solid tumor mass growing in all directions. This was achieved by allowing NSCLC A549 cells to grow in U-bottom low attachment plates, as reported by Vaidya et al. [35]. The results showed a significant decrease in the size of the 3D spheroids treated with nanoemulsion as compared to drug-free solution and control, which indicates the proficiency of the prepared nanoemulsion to penetrate deep into the tumor mass and efficiently provide tumor reduction. After lung deposition, the emulsion droplets slowly dissolve in the respiratory tract fluid and release the drug close to the tumor [66]. Passive diffusion into the cancer cells would be achieved with higher concentrations of dissolved drug in the mucus, achieving the required therapeutic effect. Both in vitro cell culture model and an ex vivo 3D spheroid model have been investigated and showed therapeutic efficacy of nanoemulsion with increased cell death and tumor regression in NSCLC.

## 6. Conclusions

The work in this study demonstrates that stable erlotinib-loaded nanoemulsion can be successfully prepared using super refined excipients via an easily scalable technique. The formulated nanoemulsion can be successfully developed as a promising approach against NSCLC to be nebulized locally into the lungs. This study also establishes hot melt extrusion as a continuous manufacturing, easily scalable technique to be used in fabrication of lipid-based nanoemulsion formulation of small molecule therapeutics, with an optimum particle size, drug loading, and good inhalability. These nanoemulsion can be nebulized for local lung delivery with sustained drug release to avoid frequent multiple dosing as a potential treatment approach for NSCLC. Moving froward, preclinical, and potentially clinical studies can be conducted to estimate the full-scale feasibility of the prepared nanoemulsion.

## Figures and Tables

**Figure 1 pharmaceutics-15-00996-f001:**
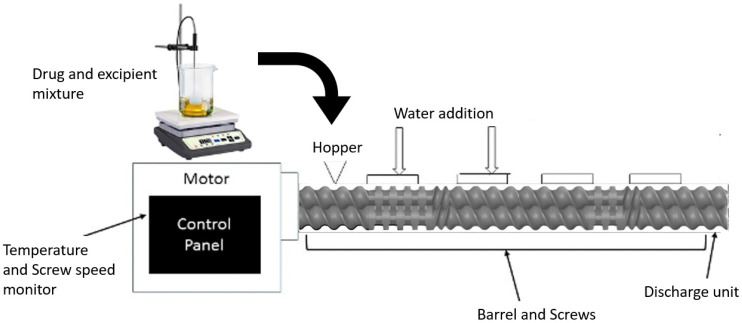
A schematic diagram representing a hot melt extruder (HME) setup for nanoemulsion formulation.

**Figure 2 pharmaceutics-15-00996-f002:**
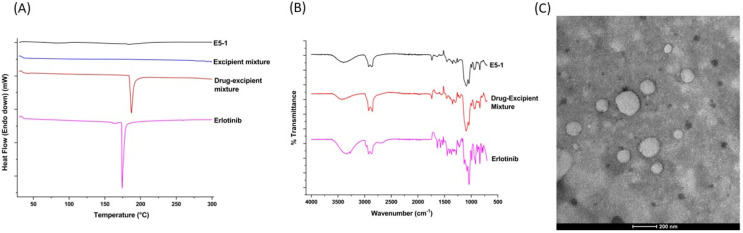
Physicochemical characterization of the nanoemulsion: diffraction scanning calorimeter thermograms (**A**) and Fourier transform infrared (**B**) of the erlotinib-free solution, drug–excipients mixture, excipients mixture (only DSC), and nanoemulsion E5-1. Transmission electron microscope (TEM) images of E5-1 nanoemulsion (**C**), with a scale bar of 200 nm.

**Figure 3 pharmaceutics-15-00996-f003:**
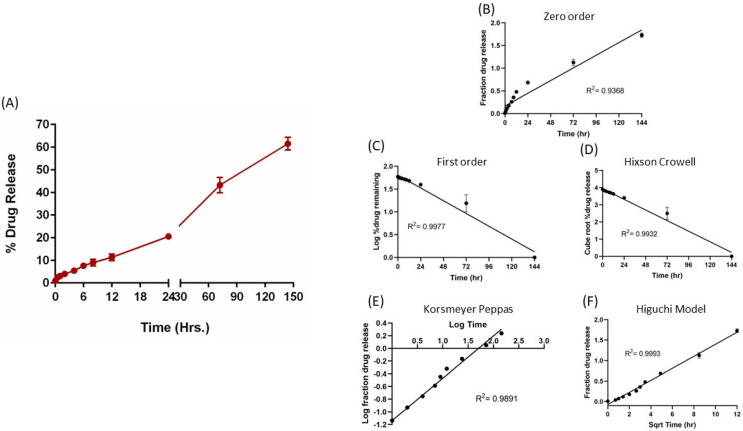
In vitro drug release profile of E5-1 nanoemulsion in phosphate buffer saline containing 1% Tween 80 at 37 °C/100 RPM using the membrane method (**A**). Drug release data fitted to various kinetics models: zero-order model (**B**), first-order model (**C**), Hixson–Crowell model (**D**), Kosmeyer–Peppas model (**E**), and Higuchi model (**F**), with the corresponding R^2^ values. Data represent mean ± SD (n = 3).

**Figure 4 pharmaceutics-15-00996-f004:**
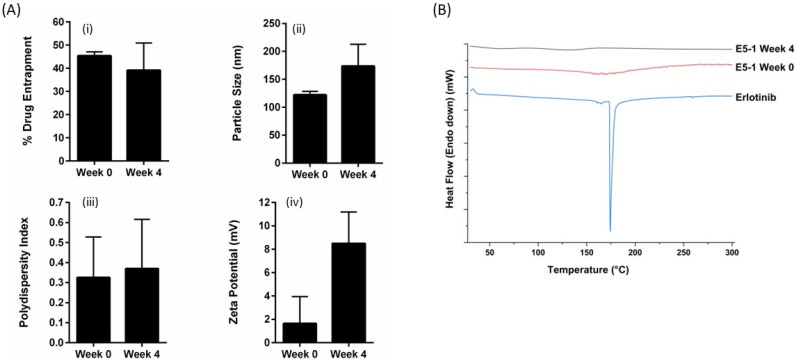
The stability study results for E5-1 nanoemulsion carried out at 40 °C/75% relative humidity over a period of 4 weeks and analyzed for the percentage of drug entrapment (i), particle size (nm) (ii), PDI (iii), zetapotential (mV) (iv) (**A**), and solid-state characteristics using differential scanning calorimeter (DSC) (**B**). Data represent mean ± SD (n = 3).

**Figure 5 pharmaceutics-15-00996-f005:**
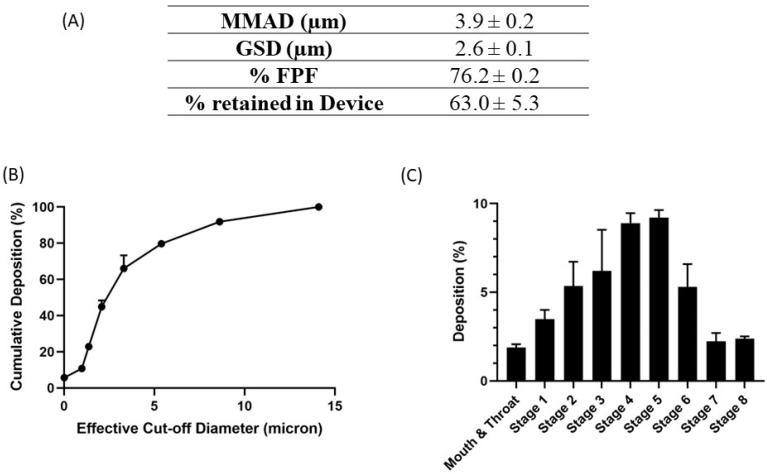
In vitro aerosolization results for E5-1 nanoemulsion, with MMAD (µm), FPF (%), and the percentage retained in the device (**A**). Cumulative mass deposition as a function of the next-generation impactor (NGI) effective cut-off diameters (**B**). Deposition pattern in the NGI (Stage 1–8) (**C**). Data represented as mean ± SD with n = 3.

**Figure 6 pharmaceutics-15-00996-f006:**
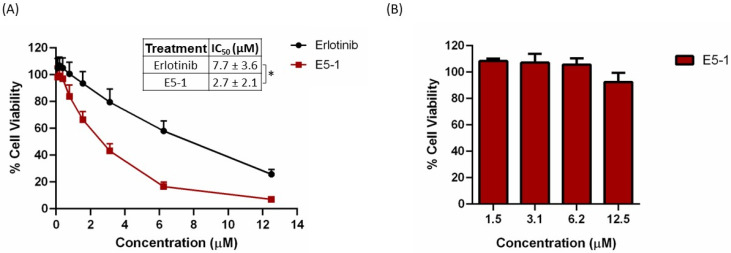
In vitro cytotoxicity study of E5-1 nanoemulsion (**A**) and drug-free nanoemulsion (**B**) in NSCLC A549 cells. Cells were treated at different concentrations (0.09 to 12.5 µM) for 72 h and cell viability was determined using MTT calorimetric assay. Data represented with mean ± SD (n = 6, for three independent trials), * *p* < 0.05.

**Figure 7 pharmaceutics-15-00996-f007:**
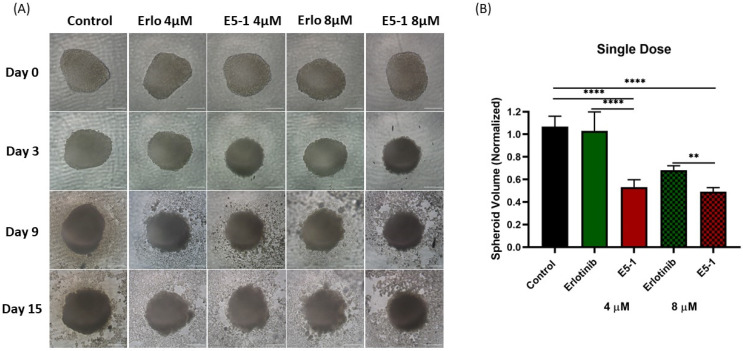
Effect of single-dose treatment on the growth of a 3D spheroid tumor of A549 cells. Spheroid images taken at day 0, 3, 9, and 15 post treatment were compared for single dose treatment with erlotinib and E5-1 nanoemulsion at 4 µM and 8 µM, Scale bar for the images represents 500 μm (**A**). The spheroids of A549 treated with E5-1 nanoemulsion showed significantly reduced spheroid volume in comparison to control and erlotinib after 15 days of treatment, as shown in the graph plots (**B**). Data represent mean ± SD (n = 6). Significance between the groups was analyzed by one-way ANOVA and Tukey’s multiple comparison test. **** *p <* 0.0001, ** *p <* 0.01.

**Figure 8 pharmaceutics-15-00996-f008:**
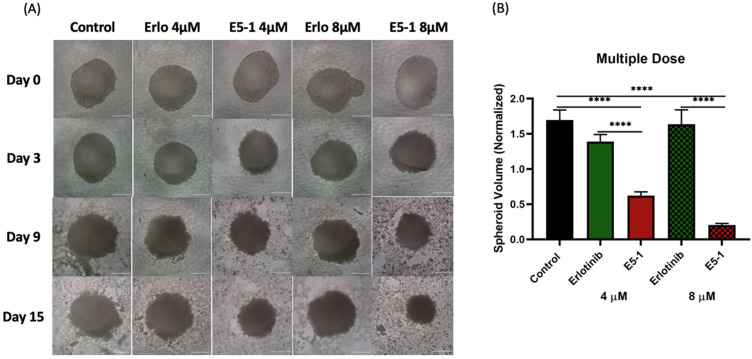
Effect of multi-dose treatment on the growth of 3D spheroid-tumor of A549 cells. Spheroid images taken after multiple dose treatment at day 0, 3, 9, and 15 for erlotinib and E5-1 nanoemulsion at 4 µM and 8 µM, Scale bar for the images represents 500 μm (**A**). The spheroids of A549 treated with E5-1 nanoemulsion showed significantly reduced spheroid volume in comparison to control and erlotinib after 15 days of treatment, as shown in the graph plots (**B**). Data represent mean ± SD (n = 6). Significance between the groups was analyzed by one-way ANOVA and Tukey’s multiple comparison test. **** *p <* 0.0001.

**Figure 9 pharmaceutics-15-00996-f009:**
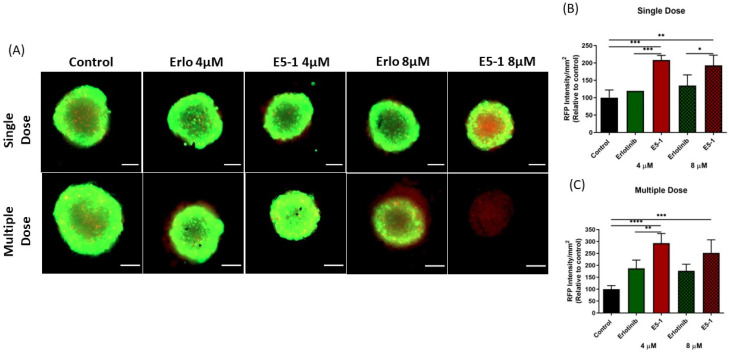
Live/dead cell assay: A549 spheroids treated with single and multiple doses were stained using the live/dead cell assay kit to determine the proportion of live and dead cells in the respective spheroids. Images were captured using an Evos-FL fluorescence microscope with a 10× objective. Spheroid stained green for live cells and red for dead cells, Scale bar for the images represents 500 μm (**A**). The fluorescent images were analyzed using ImageJ software and graphs were plotted for red fluorescence intensity signifying the proportion dead cells in the 3D spheroids for both groups, single dose (**B**) and multiple dose (**C**). Significance between the groups was analyzed by one-way ANOVA and Tukey’s multiple comparison test. Data represent mean ± SEM (n = 3). **** *p <* 0.0001, *** *p <* 0.001, ** *p <* 0.01, * *p <* 0.05.

**Figure 10 pharmaceutics-15-00996-f010:**
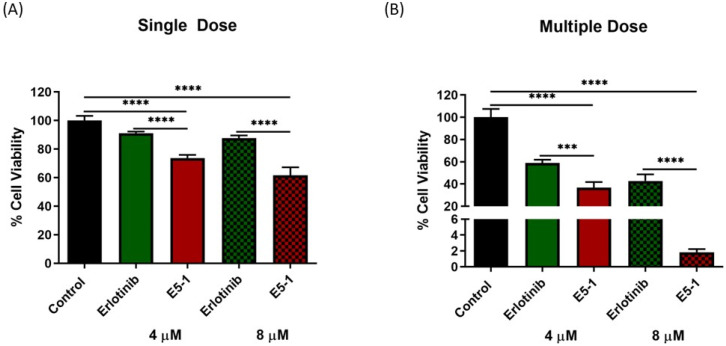
Cell viability study performed using CellTiter-Glo^®^ assay after single dose (**A**) and multiple doses (**B**). At day 15, 100 μL of medium was removed and replaced with 100 μL of Cell Titer-Glo^®^ reagent in each spheroid well and the luminescence was measured. Results indicate the percentage of cell viability after each treatment, and comparisons were made by considering the control as 100%. Significance between the groups was analyzed by one-way ANOVA and Tukey’s multiple comparisons test. Data represent mean ± SEM (n = 3). **** *p* < 0.0001, *** *p* < 0.001.

**Table 1 pharmaceutics-15-00996-t001:** Drug–excipient solubility study. Data presented as mean ± SD (n = 3).

Sample No.	Excipients	Solubility (mg/mL)
E1	L18 POG	0.1 ± 0.2
E2	Propylene glycol	2.1 ± 0.3
E3	Tween 80	7.3 ± 0.9
E4	L18 POG + Propylene glycol (1:1)	0.3 ± 0.2
E5	Tween 80 + Propylene glycol (1:1)	5.1 ± 0.6
E6	L18 POG + Tween 80 (1:1)	3.3 ± 0.4
E7	L18 POG + Propylene glycol + Tween 80 (1:1:1)	1.4 ± 0.4

**Table 2 pharmaceutics-15-00996-t002:** Drug–excipient solubility study carried out at 70 °C for 3 h. Data presented as mean ± SD (n = 3).

Sample No.	Excipients	Solubility (mg/mL)
E5	Tween 80 + Propylene glycol	8.0 ± 0.2
E6	L18 POG + Tween 80	3.5 ± 0.1
E7	L18 POG + Propylene glycol + Tween 80	1.8 ± 0.6

**Table 3 pharmaceutics-15-00996-t003:** Visual observations of emulsion stability after 24 h at room temperature. Data presented as mean ± SD (n = 3).

Sample No.	Lipid: Water	Excipients	Observation	
E5-1	1:1	Tween 80 + Propylene glycol	No separation	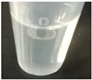
E5-2	1:2	Tween 80 + Propylene glycol	No separation	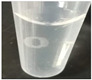
E5-3	1:3	Tween 80 + Propylene glycol	No separation	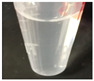
E6-1	1:1	L18 POG + Tween 80	Phase separation	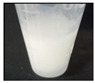
E6-2	1:2	L18 POG + Tween 80	Phase Separation	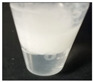
E6-3	1:3	L18 POG + Tween 80	Phase Separation	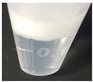

**Table 4 pharmaceutics-15-00996-t004:** Formulation characteristic table for prepared nanoemulsions using hot melt extrusion. Data presented as mean ± SD (n = 3).

Sample No.	Drug Loading (mg/mL) (% Entrapment Efficiency)	Formulation Volume (mL)	Globule Size (nm)	PDI	Zeta Potential (mV)
E5-1	4.8 ± 0.7 (55.8 ± 5.7%)	10	186.4 ± 97.6	0.231 ± 0.038	−3.2 ± 5.1
E5-2	4.4 ± 1.0 (66.7 ± 15.7%)	15	122.2 ± 58.6	0.36 ± 0.14	−0.7 ± 1.6

## Data Availability

Data available on request from the corresponding author.

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
