# Peer review of "Scalable Production and In Vitro Efficacy of Inhaled Erlotinib Nanoemulsion for Enhanced Efficacy in Non-Small Cell Lung Cancer (NSCLC)"

_pharmaceutics, 2023, doi:10.3390/pharmaceutics15030996_

Round 1

Reviewer 1 Report

1.     There are numerous grammar and formatting problems, including but not limited to: The abstract must not contain any reference. Therefore, please delete the existing reference (1) in the abstract. Add line numbers to the manuscript. “as pneumonia” (Paragraph 1, Introduction) should be changed to “such as pneumonia”. “other conventional techniques used to prepared nanoemulsion” (Paragraph 3, Introduction) should be changed to “other conventional techniques used to prepare nanoemulsion”. Please correct English and formatting issues seriously.

2.     There is a lot to improve concerning the expression and academic writing. this study is the first to(Paragraph 4, Introduction) should be changed to “to our best knowledge, this study is the first to”. Authors should be cautious to say the first. It is possible that inadequate literature retrieval causes an illusion of being the first. In the end of the Introduction Section, please add a paragraph (several sentences) to briefly introduce the purpose and main content of the study. Move Fig. S1 to this paragraph and include also efficacy research contents in it, and make the figure tell the major content of your research. It will make your manuscript more readable. The subtitles are in a mess. For instance, if Section 2 is “2. Materials and Cell Lines”, how come Section 2 contains “2.1 Experimental Setup”? Experimental setup is neither material nor cell line. In addition, why does the subtitle “Methods” have no number before it? Where does it belong? There are many subtitle problems like this. There are also mistakes on the numbering of tables. After Table 1, Table 4 appeared. Shouldn’t it be Table 2? In this sentence “Meanwhile, the emulsion prepared using propylene glycol and Tween 80 (E5) were stable” (Paragraph 2, Section 3.1 Preliminary Screening), “Meanwhile” should be changed to “In contrast”.

3.     Tentativeness is one of the requirements for academic writing. When writing results and discussion, the authors should be more tentative. There is no 100% possibility. Experiments do not prove. They imply.

4.     I‘m a little confused. Did the FT-IR results demonstrate “the absence of any chemical interactions between the excipients and drug in the E5-1 nanoemulsion” as the manuscript stated, or did they demonstrate the successful loading of drug?

5.     In Figure 7A, it appears that the spheroids on day 9 are larger than those on day 3. What may underlie this phenomenon? As the drug treatment time increases, shouldn’t the spheroid size decrease?

6.     In section 3.10. Live/Dead Cellular Assay & 3D Spheroid Cell Viability Studies, the green color comes from the fluorescent dye Calcein AM rather than GFP, and the red color are generated by ethidium homodimer III (EthD-III), as you described in the Methods section. GFP and RFP are microscope filter names indicating the excitation wavelength and emission wavelength. It is important to understand the principle of the experimental methods before you use them. Only in this way can you minimize mistakes and get close to the truth.

7.     Please consider if bi-continuous system mean unstableand drug precipitation, as you stated in Paragraph 2, Section Discussion. When using phase inversion composition (PIC) method to prepare nanoemulsion, formation of bi-continuous phase means the successful creation of small nanosized droplets.

8.     I’m a little confused by the following point: Erlotinib hydrochloride was used in your research. Is erlotinib in the hydrochloride form soluble in aqueous environment? If it is, dissolving it in lipids seems contradictory. Please explain.

Author Response

Reviewer 1

  1. There are numerous grammar and formatting problems, including but not limited to: The abstract must not contain any reference. Therefore, please delete the existing reference (1) in the abstract. Add line numbers to the manuscript. “as pneumonia” (Paragraph 1, Introduction) should be changed to “such as pneumonia”. “other conventional techniques used to prepared nanoemulsion” (Paragraph 3, Introduction) should be changed to “other conventional techniques used to prepare nanoemulsion”. Please correct English and formatting issues seriously.

Authors Response: Thank you for your suggestion. We removed the reference (1) from the Abstract and made other appropriate changes as you pointed out. We have also sent the manuscript to a native English speaker to review and make grammatical corrections.

  1. There is a lot to improve concerning the expression and academic writing. “this study is the first to” (Paragraph 4, Introduction) should be changed to “to our best knowledge, this study is the first to”. Authors should be cautious to say “the first”. It is possible that inadequate literature retrieval causes an illusion of being the first. In the end of the Introduction Section, please add a paragraph (several sentences) to briefly introduce the purpose and main content of the study. Move Fig. S1 to this paragraph and include also efficacy research contents in it, and make the figure tell the major content of your research. It will make your manuscript more readable. The subtitles are in a mess. For instance, if Section 2 is “2. Materials and Cell Lines”, how come Section 2 contains “2.1 Experimental Setup”? Experimental setup is neither material nor cell line. In addition, why does the subtitle “Methods” have no number before it? Where does it belong? There are many subtitle problems like this. There are also mistakes on the numbering of tables. After Table 1, Table 4 appeared. Shouldn’t it be Table 2? In this sentence “Meanwhile, the emulsion prepared using propylene glycol and Tween 80 (E5) were stable” (Paragraph 2, Section 3.1 Preliminary Screening), “Meanwhile” should be changed to “In contrast”.

Authors Response: Thank you for your comment, the following addition has been made to the Introduction, “to our best knowledge, this study is the first to” (Lines 124-125). We have also added a paragraph in the end of introduction to briefly introduce the purpose and main content of the research (Lines 127-135). We have also now included Fig. S1 as Fig. 1 in main body of the manuscript, and further figure numbers are thus updated. However, we have no control over the placement of figures in manuscript, as that’s done by the journal typesetters. We can request the suggested placement during galley proofing.

The subtitles have been fixed with section 2. as Materials and cell lines, and section 3. as Methods which include 3.1. as Experimental setup. The chronology of tables has been corrected, table 4 has been removed from section 3.4. as it is a part of the results. In Section 3.1. Preliminary Screening the appropriate change has been made from “Meanwhile” to “In contrast” (Line 327).

  1. Tentativeness is one of the requirements for academic writing. When writing results and discussion, the authors should be more tentative. There is no 100% possibility. Experiments do not prove. They imply.

Authors Response: Thank you so much for your comments. We have made appropriate changes in the results and discussion sections by changing the word such as, confirm, establish and proof to demonstrate, suggest, imply, and indicate.

  1.    I‘m a little confused. Did the FT-IR results demonstrate “the absence of any chemical interactions between the excipients and drug in the E5-1 nanoemulsion” as the manuscript stated, or did they demonstrate the successful loading of drug?

Authors Response: Thank you for your comment. The FT-IR studies were completed to study the chemical interactions between the drug and excipients which demonstrated no changes in the position of bands therefore absence of any chemical interaction between the excipients and drugs, this has been shown in previous studies (1,2). The following references have been added in the manuscript.

  1. In Figure 7A, it appears that the spheroids on day 9 are larger than those on day 3. What may underlie this phenomenon? As the drug treatment time increases, shouldn’t the spheroid size decrease?

Authors Response: Thank you so much for your correction. The spheroid size should decrease with increase in the treatment time which was observed for the nanoemulsion E5 at both the concentrations 4- and 8-µM. We have made the appropriate corrections and updated the representative spheroid images in the manuscript for better representation of the treatments (Fig. 7A). However, for plain drug treatments this spheroid size reduction was not observed, potentially due to poor water solubility of Erlotinib which causes recrystallization of the drug when more drug is added in multiple treatments, thus resulting in limited cell internalization. Moreover, 3D spheroids have large number of cells in the core which continually grow, therefore we observed an increase in the spheroid size from Day 3 to Day 15 for plain drug, which is a similar phenomenon observed in previous studies (3,4).

  1. In section 3.10. Live/Dead Cellular Assay & 3D Spheroid Cell Viability Studies, the green color comes from the fluorescent dye Calcein AM rather than GFP, and the red color are generated by ethidium homodimer III (EthD-III), as you described in the Methods section. GFP and RFP are microscope filter names indicating the excitation wavelength and emission wavelength. It is important to understand the principle of the experimental methods before you use them. Only in this way can you minimize mistakes and get close to the truth.

Authors Response: Thank you for your comment. Respective edits are made in the manuscript to clearly outline the experimental procedure (Line 445).

  1. Please consider if bi-continuous system mean “unstable” and “drug precipitation”, as you stated in Paragraph 2, Section Discussion. When using phase inversion composition (PIC) method to prepare nanoemulsion, formation of bi-continuous phase means the successful creation of small nanosized droplets.

Authors Response: Thank you for your comment. We meant stability concerns to the bi-continuous phase of the nanoemulsion due to water dilution (5). We have made appropriate changes in the manuscript (Lines 505-506).

  1. I’m a little confused by the following point: Erlotinib hydrochloride was used in your research. Is erlotinib in the hydrochloride form soluble in aqueous environment? If it is, dissolving it in lipids seems contradictory. Please explain.

Authors Response: Thank you for your comment. Yes, the drug used is salt form of erlotinib, however, erlotinib hydrochloride is very slightly soluble in water (0.00891 mg/mL) (3,6). Therefore, dissolving it in lipids would make sense as a means for formulation development to improve low water solubility issues for improved drug delivery.

Reviewer 2 Report

his manuscript reports the production and in vitro efficacy of inhaled erlotinib nano-emulsion to treat resistant NSCLC. In general this is an interesting manuscript. However, there are some issues that the reviewer would like to highlight to the authors. 

1) The title should reflect the reported results or conducted experiments. For example, scalable and resistant word may not be suitable in the title because there are no such data provided in this manuscript. Please kindly re-word the title or supply the data for it.

2) Please provide the methods to eliminate undissolved drugs and excipients prior to the further analysis. The authors need to ensure that the undissolved drugs and excipients have been totally eliminated in results.

3) Any pre-treatment conducted with the sample prior to the TEM imaging? Please include it, if any.

4) Any pre-treatment involved in the in vitro release experiment before subjected to UPLC analysis?

5) Please include the aerosol data of erlotinib loaded nanoemulsion before and after subjected to stability test. The same data are needed for in vitro cytotoxicity (A549 cells).

6) The authors should include the effect of storage condition towards the efficacy of erlotinib loaded nanoemulsion.

Author Response

Reviewer 2

This manuscript reports the production and in vitro efficacy of inhaled erlotinib nano-emulsion to treat resistant NSCLC. In general this is an interesting manuscript. However, there are some issues that the reviewer would like to highlight to the authors. 

  1. The title should reflect the reported results or conducted experiments. For example, scalable and resistant word may not be suitable in the title because there are no such data provided in this manuscript. Please kindly re-word the title or supply the data for it.

Authors Response: Thank you so much for your suggestion. The scalable word has been used in the title as the technique used to prepare these nano-emulsions is a single-step continuous manufacturing technique which has been identified by the FDA as being scalable (7–9). This has been explained with appropriate citation in the manuscript (Lines 91-93).

To address the comment about resistant NSCLC, the title of the manuscript has been updated to “Scalable Production and In-vitro Efficacy of Inhaled Erlotinib Nano-emulsion for Enhanced Efficacy in Non-small cell lung cancer (NSCLC)”.

  1. Please provide the methods to eliminate undissolved drugs and excipients prior to the further analysis. The authors need to ensure that the undissolved drugs and excipients have been totally eliminated in results.

Authors Response: Thank you so much for your comment. The undissolved drug and excipients were removed with the help of centrifugation after the emulsion was collected from the extruder. The method in the manuscript has been updated accordingly (Lines 198-200).

  1. Any pre-treatment conducted with the sample prior to the TEM imaging? Please include it, if any.

Authors Response: Thank you for your comment. There was no pre-treatment performed before taking the TEM images. The nanoemulsion was placed on a formvar-carbon-coated copper grid and imaged as mentioned in the method section.

  1. Any pre-treatment involved in the in vitro release experiment before subjected to UPLC analysis?

Authors Response: Thank you for the comment. There was no pre-treatment involved before analyzing the samples using UPLC. The collected release samples were diluted with the same volume of acetonitrile and analyzed using the UPLC as mentioned in the method section.

  1. Please include the aerosol data of erlotinib loaded nanoemulsion before and after subjected to stability test. The same data are needed for in vitro cytotoxicity (A549 cells).

Authors Response: Thank you so much for your comment. The results of the stability study in this project showed that drug entrapment efficacy, particle size, PDI, zeta potential, and drug polymorphism for the nanoemulsion remained consistent from week 0 to week 4, confirming their stability over period of 28 days in accelerated conditions. From these results, we can be certain about consistency in the aerosolization behavior and in vitro efficacy of stored formulations. For the current study, the preliminary results demonstrated the scalable production of inhaled nanoemulsions for enhanced efficacy in NSCLC. Ensuring stability and efficacy of stored nanoparticle-based formulations is an exciting, however, a likely new project.

  1. The authors should include the effect of storage condition towards the efficacy of erlotinib loaded nanoemulsion.

Authors Response: Thank you so much for your comment. The results of the stability study showed drug entrapment efficacy, particle size, PDI, zeta potential and drug polymorphism for the nanoemulsion remained consistent over period of 4 weeks in accelerated conditions (40ºC/75% RH), confirming their stability. From these results, we can imply in-vitro efficacy can be maintained, as reported in previous studies (3,10,11). We would like to include your suggestion of testing the efficacy of erlotinib loaded nanoemulsion after storage in our future studies.

Reviewer 3 Report

The paper “Scalable Production and In-vitro Efficacy of Inhaled Erlotinib Nano-emulsion for Enhanced Efficacy in Resistant NSCLC” describes an early-stage development of new carrier for erlotinib intended for the treatment of NSCLC by the pulmonary route. The paper is clear and well written. The data are appropriately presented and the statistical analysis is good. However some methods and the absence of some critical controls are questionable in the manuscript. Some conclusions (and the title) are, according to me, not supported by the data of the manuscript which should be published in Pharmaceutics after major reviewing.

General comment:

-       The title should be revised as there are no data in the manuscript supporting enhanced efficacy in resistant NSCLC.

-       In the introduction, the authors only mention chemotherapy and radiotherapy as the currently available therapeutic protocols of NSCLC, which is incomplete. They should also mentioned the use of surgery, targeted therapy (such as TKIs) and immunotherapy, which are important parts of the standard of care treatment of NSCLC.

-       The authors define erlotinib (and TKI) as chemotherapy whereas it is commonly defined in the field as targeted therapy (DOI: 10.3390/cancers13184705; DOI: 10.1038/s41392-019-0099-9).

-       Drug loading – fine particle dose – anticipated doses to be inhaled. It is not clear whether nebulization of the nanoemulsion would reach efficient concentrations in the lungs and in the tumor site. This should be discussed.

-       The numbers of the Tables should be revised.

-       The methods should not be redescribed in the Results section.

Comments:

-       The reference of the UPLC column should be indicated (reference, supplier). The method should not be redescribe in the Results section.

-       The authors refer to a miscibility study whereas solubility data are presented (Tables 1 & 2). What is determined exactly? It should be clarified.

-       Determination of the drug loading and % entrapment efficiency. How was the O phase separated from the W phase in the nanoemulsion? The Section 3.2. only refer to centrifugation to remove undissolved drug. Do the authors expect to have undissolved drug in the nanoemulsion compositions? This should be clarified.

-       Dissolution test. A solution of erlotinib should be used as a control of the method as it involves dialysis through a membrane. It is not clear what the aim of such a test is as the in vitro – in vivo correlation is highly questionable for inhaled products. Why did the authors conduct such a test?

-       The Fine Particle Dose (or mass) and its fraction related to the emitted dose is defined by the pharmacopeia as the drug dose or fraction of particle below 5 µm. Why do the authors define the FPF as particles fraction below 5.39 µm?

-       The authors define the MTT test as a cytotoxicity test, which his incorrect. The MTT test is based on the metabolism of cells, i.e. it calculates the fraction of cells metabolically alive. The effect of a drug on the cells evaluated using a MTT test can be an antiproliferative effect which can be a cytotoxic effect but also antiadherent, cytostatic, … effects. The term antiproliferative test is more appropriate. It should be modified throughout, including conclusions made from such a test.

-       In the DSC analysis, was the drug-excipient mixture prepared in the same proportion as in the nanoemulsion?

-       In Section 3.4, in the FT-IR discussion, the authors refer to the absence of any “chemical” interaction. The term covalent linkage might be more appropriate.

-       Stability studies. Were the stab studies perform on the nanoemulsion or the free-dried product? I assume it is on NE as it includes DLS measurement. But if it on a liquid, how did the authors perform DSC analysis (for solid-state characterization)? Removing liquid NE from the stab chamber, drying the product by free-drying and conduct the DSC analysis (as it is described in the DSC method session) makes no sense to investigate stability. It should be discussed and clarified.

-       I do not agree with the following conclusion “These results ascertain that the enhanced cytotoxic effect of E5-1 nanoemulsion against NSCLC is attributed mainly to improved nanoemulsion-medicated drug accumulation inside the cells and subsequently, enhanced therapeutic activity.” There are no data in the manuscript supporting any improved drug accumulation inside the cells.

-       3D tumours treated multiple times.

o   What is the rationale behind the frequency of exposure / treatment (i.e. every 3 days)? What is the difference between multiple exposures and exposure at higher concentrations since erlotinib remains manly entrapped in the NE within 3 days (~50% erlotinib released with 72h according to Fig 2A).

o   How do the authors explain the lower activity of erlonitib on A549 spheroids following multiple exposures compared with single exposure (Gig 6 & 7, ~0.7 and 1.2 at 8 µM, respectively)?

-       In the discussions: “Passive diffusion into the cancer cells would be achieved with higher concentrations of dissolved drug in the mucus, achieving the required thera-peutic effect.” Are high concentrations in the mucus really expected, considering (i) the low dissolution rate over time and (ii) the elaborated mechanism of defense of the lungs against foreign particles (eg mucociliairy escalator, absorption to systemic circulation…)?

Author Response

Reviewer 3

The paper “Scalable Production and In-vitro Efficacy of Inhaled Erlotinib Nano-emulsion for Enhanced Efficacy in Resistant NSCLC” describes an early-stage development of new carrier for erlotinib intended for the treatment of NSCLC by the pulmonary route. The paper is clear and well written. The data are appropriately presented, and the statistical analysis is good. However, some methods and the absence of some critical controls are questionable in the manuscript. Some conclusions (and the title) are, according to me, not supported by the data of the manuscript which should be published in Pharmaceutics after major reviewing.

General comment:

  1. The title should be revised as there are no data in the manuscript supporting enhanced efficacy in resistant

Authors Response: Thank you so much for your comment. The title of the manuscript has been updated to “Scalable Production and In-vitro Efficacy of Inhaled Erlotinib Nano-emulsion for Enhanced Efficacy in Non-small cell lung cancer (NSCLC)”.

  1. In the introduction, the authors only mention chemotherapy and radiotherapy as the currently available therapeutic protocols of NSCLC, which is incomplete. They should also mention the use of surgery, targeted therapy (such as TKIs) and immunotherapy, which are important parts of the standard of care treatment of NSCLC.

Authors Response: Thank you so much for your comment. The following additions have been made in the introduction section with appropriate references (Lines 61 through 72).

  1. The authors define erlotinib (and TKI) as chemotherapy whereas it is commonly defined in the field as targeted therapy (DOI: 10.3390/cancers13184705; DOI: 10.1038/s41392-019-0099-9).

Authors Response: Thank you so much for your comment. We have clarified the sentence in the introduction section (Line 72) as follows, “Tyrosine kinase inhibitors (TKIs) have been at the forefront of chemotherapeutic advances as targeted therapy against NSCLC”.

  1. Drug loading – fine particle dose – anticipated doses to be inhaled. It is not clear whether nebulization of the nanoemulsion would reach efficient concentrations in the lungs and in the tumor site. This should be discussed.

Authors Response: Thank you so much for your comment. The aerosolization study indicates that >75% of the emitted dose will reach the respirable region of the lungs. However, we cannot predict the dose required for inhalation therapy against NSCLC from the in-vitro studies performed in this project alone. This is a preliminary project with the objective of preparing inhalable erlotinib loaded nanoemulsions using scalable technique for enhanced in-vitro efficacy against NSCLC. For future studies, in a separate project we would like to evaluate the in-vivo efficacy in a lung cancer animal model, by doing so we can then predict human equivalent dose and discuss about the effective dose for inhalation therapy to target NSCLC. In addition, we did a PubMed search for inhaled erlotinib to treat NSCLC and found there are no publications that can elucidate the human dose.

  1. The numbers of the Tables should be revised.

Authors Response: Thank you so much for your comment. The listing of tables has been corrected. Table 4 has been removed from Section 3.4., as it is a part of the results.

  1. The methods should not be redescribed in the Results section.

Authors Response: Thank you so much for your comment. The methods used for performing the experiments have been removed from the Results section.

Comments:

  1. The reference of the UPLC column should be indicated (reference, supplier). The method should not be redescribe in the Results section.

Authors Response: Thank you so much for your comment. The method for UPLC analysis has been updated with the supplier information (Lines 164-165), the method has been removed from the Results section.

  1. The authors refer to a miscibility study whereas solubility data are presented (Tables 1 & 2). What is determined exactly? It should be clarified.

Authors Response: Thank you so much for your comment. In the study, the term miscibility has been used to describe the homogenous mixture of the excipients, whereas the solubility has been used to describe the drug solubility in the excipient mixture. The method section 3.3.1. Drug-excipient miscibility has been changed to Drug-excipient solubility (Line 170). In addition, table captions for Table 1 & 2 have been updated accordingly.

  1. Determination of the drug loading and % entrapment efficiency. How was the O phase separated from the W phase in the nanoemulsion? The Section 3.2. only refer to centrifugation to remove undissolved drug. Do the authors expect to have undissolved drug in the nanoemulsion compositions? This should be clarified.

Authors Response: Thank you so much for your valuable comment. The Section 3.3. (earlier Section 3.2) describes the method for testing the solubility of the drug in the lipid excipients, there was no water phase present for this study. The purpose of this study was to understand the drug solubility in different lipid excipients and thereby selecting the lipid with maximum drug solubility. The tubes were centrifuged at high speed of 21,000×g to ensure the undissolved drug was pelletized at the bottom of the tube so the drug dissolved in the lipid mixture can be analyzed. The effect of water phase on the emulsion was tested in the next study Section 3.3.2. Dispersibility of the Nanoemulsion.

  1. Dissolution test. A solution of erlotinib should be used as a control of the method as it involves dialysis through a membrane. It is not clear what the aim of such a test is as the in vitro – in vivo correlation is highly questionable for inhaled products. Why did the authors conduct such a test?

Authors Response: Thank you for your valuable suggestion. Erlotinib hydrochloride has very poor solubility in water (0.00891 mg/mL), therefore it is not feasible to run a dissolution study for the plain drug, as the drug concentration in the collected samples would be too low for quantification (12). Dissolution study was carried out for the Erlotinib loaded nanoemulsion in 0.1% tween 80 PBS solution which contains salts and surfactant to simulate the lung fluid. This study gives an understanding of the release profile of the drug once deposited in the lung, similar studies have been performed in the previous projects involving inhaled nanocarriers (13,14).

  1. The Fine Particle Dose (or mass) and its fraction related to the emitted dose is defined by the pharmacopeia as the drug dose or fraction of particle below 5 µm. Why do the authors define the FPF as particles fraction below 5.39 µm?

Authors Response: Thank you for your valuable comment. The Fine particle fraction is defined as the fraction of emitted dose deposited in the stage 3 to 8 (15). The cut off diameter of stage 3 at 15 L/min flow rate on the Next Generation Impactor is equivalent to 5.39 µm, therefore the particle dose with aerodynamic diameter less than 5.39 µm are defined as Fine Particle Dose. Mass median aerodynamic diameter (MMAD) is the diameter at which 50% of the particles by mass are larger and 50% are smaller than median, and should be <5 µm for an optimal deep lung deposition. These have been mentioned in the methods section and reported in previous studies (10,11,16)

  1. The authors define the MTT test as a cytotoxicity test, which is incorrect. The MTT test is based on the metabolism of cells, i.e. it calculates the fraction of cells metabolically alive. The effect of a drug on the cells evaluated using a MTT test can be an antiproliferative effect which can be a cytotoxic effect but also antiadherent, cytostatic, … effects. The term antiproliferative test is more appropriate. It should be modified throughout, including conclusions made from such a test.

Authors Response: Thank you for your valuable comment. As per the manufacturer guidelines, MTT assay is colorimetric assay used to measure cellular metabolic activity, which is an indicator of cell viability, proliferation, and cytotoxicity. In this project we have used this assay to determine cell viability for cytotoxicity assessment. This assay has been extensively used and published reports for quantifying cell viability or cytotoxicity (3,11,17,18).

  1. In the DSC analysis, was the drug-excipient mixture prepared in the same proportion as in the nanoemulsion?

Authors Response: Thank you so much for your valuable comment. Yes, for DSC analysis the drug-excipient mixture was prepared in same proportion as the drug present in the nanoemulsion.

  1. In Section 3.4, in the FT-IR discussion, the authors refer to the absence of any “chemical” interaction. The term covalent linkage might be more appropriate.

Authors Response: Thank you so much for your comment. I agree with your suggestion, the covalent linkage is more appropriate for FTIR studies and has been modified in the manuscript (Lines 357-358, and Line 512).

  1. Stability studies. Were the stab studies performed on the nanoemulsion or the free-dried product? I assume it is on NE as it includes DLS measurement. But if it on a liquid, how did the authors perform DSC analysis (for solid-state characterization)? Removing liquid NE from the stab chamber, drying the product by free-drying and conduct the DSC analysis (as it is described in the DSC method session) makes no sense to investigate stability. It should be discussed and clarified.

Authors Response: Thank you for your comment. The stability studies were performed for liquid nanoemulsion after storing at accelerated condition (40ºC/75%RH) over a period of 4 weeks. The liquid nanoemulsion showed consistent drug entrapment efficacy, particle size, PDI, and zeta potential. Thereafter, to ensure that the drug did not recrystallize and remained in amorphous state we performed DSC for freeze dried samples. Results indicated the drug did not recrystallize and remained in the amorphous state inside the nanoemulsion during the storage period. This information has been updated in the manuscript (Line 385).

  1. I do not agree with the following conclusion “These results ascertain that the enhanced cytotoxic effect of E5-1 nanoemulsion against NSCLC is attributed mainly to improved nanoemulsion-medicated drug accumulation inside the cells and subsequently, enhanced therapeutic activity.” There are no data in the manuscript supporting any improved drug accumulation inside the cells.

Authors Response: Thank you so much for your comment. Nanoemulsion have been used to deliver anti-cancer drug with enhanced cellular uptake in previous studies (19,20). Therefore, the enhanced cytotoxic effect of nanoemulsions can be expected due to drug accumulation inside the cells. The statement has been modified to “The enhanced cytotoxic effect of E5-1 nanoemulsion against NSCLC can be attributed to improved nanoemulsion-medicated drug accumulation inside the cells as lipid and surfactant components of the nanoemulsion can improve cell membrane permeability and subsequently, enhanced therapeutic activity” (Lines 414-416).

  1. 3D tumors treated multiple times.
    1. What is the rationale behind the frequency of exposure / treatment (i.e. every 3 days)? What is the difference between multiple exposures and exposure at higher concentrations since erlotinib remains manly entrapped in the NE within 3 days (~50% erlotinib released with 72h according to Fig 2A).

Authors Response: Thank you so much for your comment. In 3D tumor spheroids studies frequency of multiple dose treatments after every 3 days is a standard protocol followed in various published works (13,21). The purpose of multiple dose treatment is to see if repeated treatments would further inhibit the growth of spheroid volume over time. Also, after each 3-day interval, half of the previous treatments are removed and replenished with the fresh treatments. This is followed to maintain a certain concentration in the 3D spheroids, thus constantly exposing the spheroids with the treatments. 3D spheroids studies are conducted to compare the single- and multiple-dose regimen to analyze whether one favors over the other. In addition, 3D spheroids do not account for the amount of drug cleared or removed from the body, therefore it simply creates a bridge between in-vitro and in-vivo studies. In-vivo animal studies are to be conducted in the future to fully elucidate the appropriate dose and frequency for optimal therapeutic efficacy.

  1. How do the authors explain the lower activity of erlotinib on A549 spheroids following multiple exposures compared with single exposure (Fig 6 & 7, ~0.7 and 1.2 at 8 µM, respectively)?

Authors Response: Thank you so much for your comment. The 3D spheroids treated with nanoemulsion at concentration of 8-µM showed a reduction in the volume after multiple dose treatment as compared to single dose. However, we did not see the same results for plain drug, in contrary the multiple dose treatment for plain drug had larger spheroid volume as compared to single dose. This can be attributed to poor water solubility of Erlotinib which causes recrystallization of the drug when more drug is added in multiple treatments, therefore resulting in limited cell internalization. On other hand, 3D spheroids have large number of cells in the core which continually grow therefore we observed an increase in the spheroid volume from Day 3 to Day 15 for plain drug, also observed in previous studies (3,4).

  1. In the discussions: “Passive diffusion into the cancer cells would be achieved with higher concentrations of dissolved drug in the mucus, achieving the required therapeutic effect.” Are high concentrations in the mucus really expected, considering (i) the low dissolution rate over time and (ii) the elaborated mechanism of defense of the lungs against foreign particles (eg mucociliary escalator, absorption to systemic circulation…)?

Authors Response:

  • Thank you for your comments. The sustained drug release from the prepared nanoemulsions will affect the dose volume and dosing frequency. However, the effective concentration will reach in the lungs using the nanoemulsions confirmed with in-vitro aerosolization studies which will show better therapeutic efficacy as compared to plain drug indicated by the in-vitro cytotoxicity and an ex-vivo 3D-spheroid studies.
  • Thank you for your perspective. I agree with your comment that lungs defense against foreign objects such as mucociliary clearance will clear the nanoemulsions from the lungs with the turnover period of 24-48 hours. However, the small size of the nanoemulsion (100 – 200 nm) will enable mucus penetration and avoid Mucociliary clearance (22).

Round 2

Reviewer 1 Report

About the FTIR results, please delete the words “successful encapsulation of drug in the nanoemulsions with” (Line 364, Paragraph 2, Section 3.4. Solid State Characterization). The reasons are as follows: According to the detailed description on FTIR results in the supplementary materials of the reference you provided (DOI: 10.1016/j.ijbiomac.2018.10.181), to demonstrate the successful drug encapsulation, difference should exist between the E5-1 FTIR spectrum and the Drug-Excipient Mixture spectrum. However, the manuscript text did not involve this description.

Author Response

Comment: About the FTIR results, please delete the words “successful encapsulation of drug in the nanoemulsions with” (Line 364, Paragraph 2, Section 3.4. Solid State Characterization). The reasons are as follows: According to the detailed description on FTIR results in the supplementary materials of the reference you provided (DOI: 10.1016/j.ijbiomac.2018.10.181), to demonstrate the successful drug encapsulation, difference should exist between the E5-1 FTIR spectrum and the Drug-Excipient Mixture spectrum. However, the manuscript text did not involve this description.

Author response: Thanks for your comment. We have now removed the statement, as suggested by the reviewer (Section 4.4, Line 358-359).

Reviewer 3 Report

I recommend the publication of the paper. 

Author Response

Comment: I recommend the publication of the paper. 

Author response: Thanks for your extensive review of the manuscript.